# Characterizing a Wedged Chalk Prospect in the Danish Central Graben Using Direct Probabilistic Inversion

Kenneth Bredesen [1,*] , Ian Herbert [2], Florian Smit [1], Ask Frode Jakobsen [2], Peter Frykman [1] and Anders Bruun [2]

[1] Geological Survey of Denmark and Greenland (GEUS), 1350 Copenhagen, Denmark; fs@geus.dk (F.S.); pfr@geus.dk (P.F.)

[2] Qeye, 2100 Copenhagen, Denmark; ian@qeye-labs.com (I.H.); afj@qeye-labs.com (A.F.J.); ab@qeye-labs.com (A.B.)

\* Correspondence: kenb@geus.dk

**Abstract:** A novel direct probabilistic inversion using seismic pre-stack data as input to characterize a wedged chalk reservoir prospect was demonstrated from the Upper Cretaceous unit, Danish North Sea. The objective was to better resolve the lateral extent and pinch-out of the chalk prospect in a frontier exploration setting and compare the results with a more traditional deterministic inversion and geostatistical reservoir modeling. The direct probabilistic inversion results provided additional reservoir insights that were challenging to obtain from the more traditional workflows and are also more flexible for associated uncertainty assessments. Hence, this study demonstrates the usefulness of such direct probabilistic inversions even with suboptimal data availability.

**Keywords:** reservoir geophysics; Direct Seismic Inversion; interpretation; Bayesian inference problem; Danish Central Graben





## 1. Introduction

In seismic reservoir characterization, there is a continuous push toward improving quantitative interpretation accuracy through innovative inversion tools [1]. Seismic reflection data are the result of elastic contrasts between different layers, whereas, for reservoir characterization, we are interested in the reservoir parameters within the individual layers. Transforming the relative interface properties (two-way travel time, amplitude, reflectivity) from seismic data into the absolute layer properties (acoustic impedance, Vp/Vs, lithology, porosity, fluid) represents the underlying objective for seismic inversion workflows. Going from interface to layer properties via seismic inversion can provide more detailed and high-resolution subsurface images for reservoir characterization by better balancing the seismic frequencies and compensating for geophysical artefacts, such as seismic tuning.

Traditionally, seismic prediction of lithology and fluid has been defined as a two-step approach. First, the seismic data are inverted to elastic properties, and then the reservoir properties are estimated through a rock physics inversion or lithology/facies classification step [2–5]. More recently, several new inversion methods have combined these two steps into a one-step (or direct) inversion to be able to integrate more geological and geophysical spatial information in a Bayesian (or probabilistic) framework (Figure 1) [6–8]. Moreover, because there is more than one possible solution to practical inversion problems (i.e., non-uniqueness), using probabilistic inversion methods allows for assessing a broader range of solutions, and thereby obtaining better control of associated uncertainties in the risk assessments.

In this paper, a novel direct probabilistic inversion (DPI) method using seismic amplitude vs. offset (AVO) data is presented to characterize a wedged chalk prospect in the Danish Central Graben (Figure 2). The prospect is located on the northern flank of the Pollernes Ridge and is a potential stratigraphic trap within the Upper Cretaceous to Earliest Paleocene Chalk Group, where highly porous reservoir facies (Ekofisk and Tor formations)

are wedged between a basal seal (low-porosity Hod Formation) and upper seal (Paleocene clays) (Figure 2). The objective of this study was to test whether the DPI results could better resolve the lateral extent and pinch-out of the chalk prospect. We first review the DPI method and study area before presenting the DPI setup and results, which are compared with the acoustic impedance and Vp/Vs properties derived from a deterministic AVO inversion. Then we compare the DPI results that are driven by seismic AVO data with a more traditional geostatistical reservoir model that is mainly based on nearby well log data. Finally, some conclusions are stated.

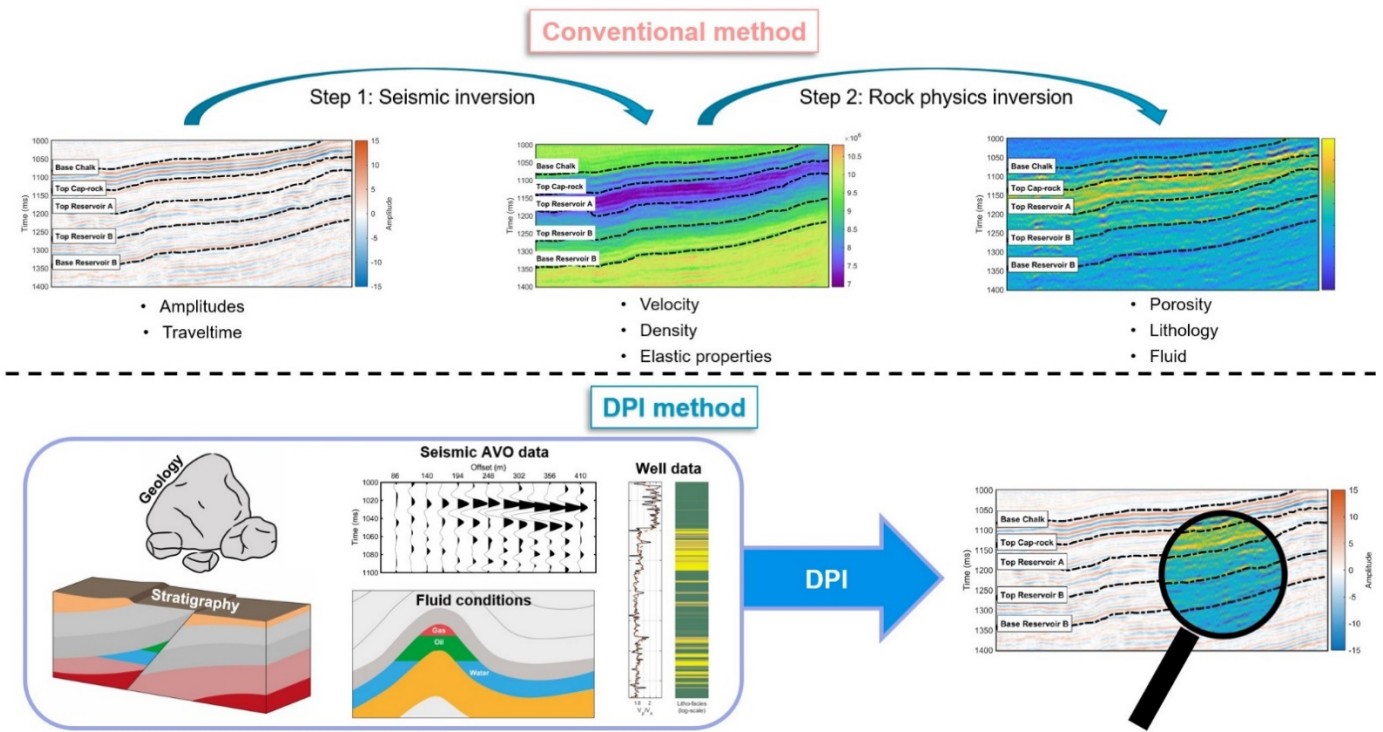

**Figure 1.** Schematic illustration of seismic inversion by the conventional two-step method (**upper**) and a one-step (direct) probabilistic inversion method (**lower**).

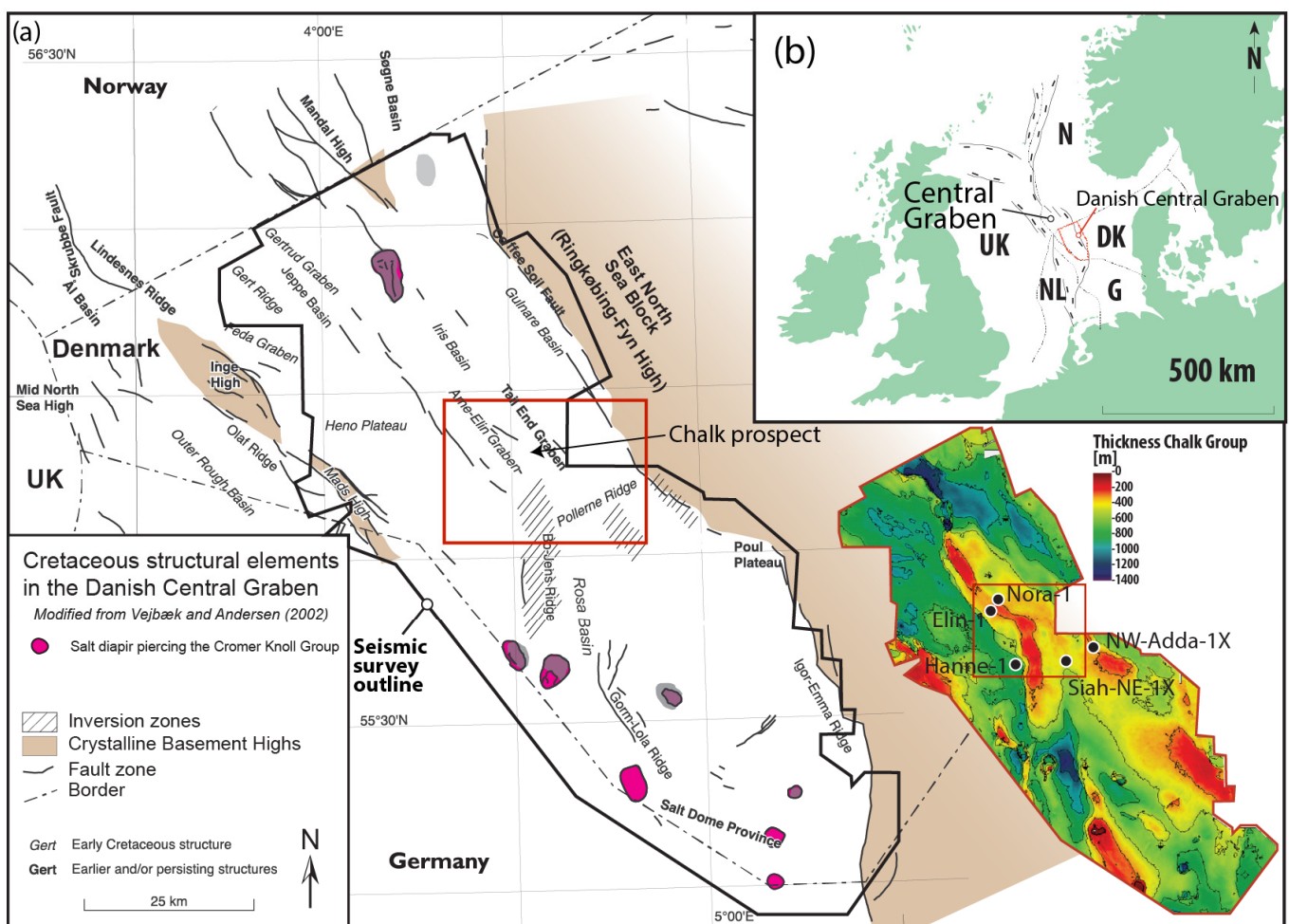

**Figure 2.** Map of the Danish Central Graben with the location of the different case studies. (**a**) Map with structure elements during the Cretaceous, adapted with permission from Reference [9] under Creative Commons Attribution 4.0 License. (**b**) Location of the study area within NW Europe.

## 2. Method: Direct Probabilistic Inversion

This section provides a description of the direct probabilistic inversion (DPI) tool, as described in Hansen and Jakobsen [10] and Mutual et al. [11]. The DPI is a single-step inversion process that inverts pre-stack seismic data directly for facies or litho-fluid classes through integration of seismic AVO data, well logs and geological information of spatial facies distribution. The seismic AVO inverse problem can be formulated as a Bayesian inference problem, where the Bayes rule can be written as follows [12]:

$$\sigma(\boldsymbol{m}) = c\rho(\boldsymbol{m})L(\boldsymbol{d_{obs}} - g(\boldsymbol{m})), \tag{1}$$

where $\boldsymbol{m}$ represents the subsurface model parameter (e.g., facies, porosity, saturation, etc.). Here, the information about $\boldsymbol{m}$ is described by a probability density function (pdf). In the initial state of the inference, before taking seismic data into consideration, the information is described by the prior pdf, $\rho$. The prior pdf is updated with the information provided by seismic AVO data via the likelihood function, $L$, which measures, in terms of probability, the misfit between forward modeled $g(\boldsymbol{m})$ and measured seismic AVO data, $\boldsymbol{d_{obs}}$. The $c$ is a normalization constant, and the Bayesian (or posterior) pdf, $\sigma$, represents the updated state of inference of the subsurface model parameters, $\boldsymbol{m}$, assimilating the prior, the AVO data and the forward modeling, $g$. The problem of non-uniqueness disappears when solving for a pdf and resolves many of the associated problems when interpreting standard inversion attributes, including correctly propagating uncertainty and spatial dependencies.

In general, the posterior pdf cannot be solved analytically, and the inverse problem must therefore be approached by brute-force sampling or some approximation, with DPI taking the latter approach.

Markov chain Monte Carlo sampling methods [12] can be demonstrated to provide an ensemble of samples which will converge on the posterior pdf. However, efficiency is very problem-dependent and may be difficult to achieve. In addition, required sampling density in a high dimensional model space (such as seismic volumes) means that these methods, in general, are computationally demanding and time-consuming. However, a number of approximations can be made to make the inversion problem computationally manageable by solving for the pointwise posterior, using a localized and approximated likelihood model [13]. The DPI method has very few requirements on spatial facies model, rock physics model, seismic forward model, etc. In the following we use a first-order Markov process to model key geologic information of facies [14]. This is used to encode geological rules from prior knowledge and statistics from well data directly into the inversion problem. The statistical rock physics (RPM) prior model for each facies is implemented as a Gaussian pdf in the elastic domain estimated from well logs (or prior rock physics knowledge/modeling). Hence, information such as layer ordering, facies thicknesses, mean and standard deviations of elastic properties for each facies, intra property correlations and any meaningful data that can be described as a probability can be implemented into the inversion problem. Combining the rock physics likelihood model with a seismic convolutional AVO forward model from elastic properties [15,16] of our defined facies yields a localized likelihood model that can answer how likely a localized piece of seismic is centered on a given facies window. Adding prior information from different domains directly into the inversion can potentially resolve beyond the seismic bandwidth. See Hansen and Jakobsen [10] for more conceptual DPI details. A schematic workflow of the DPI inversion is shown in Figure 3, where the inputs are given as the prior framework (stratigraphical/geological information and well data) and seismic AVO data, whereas the main output is the marginal (pointwise) posterior probability.

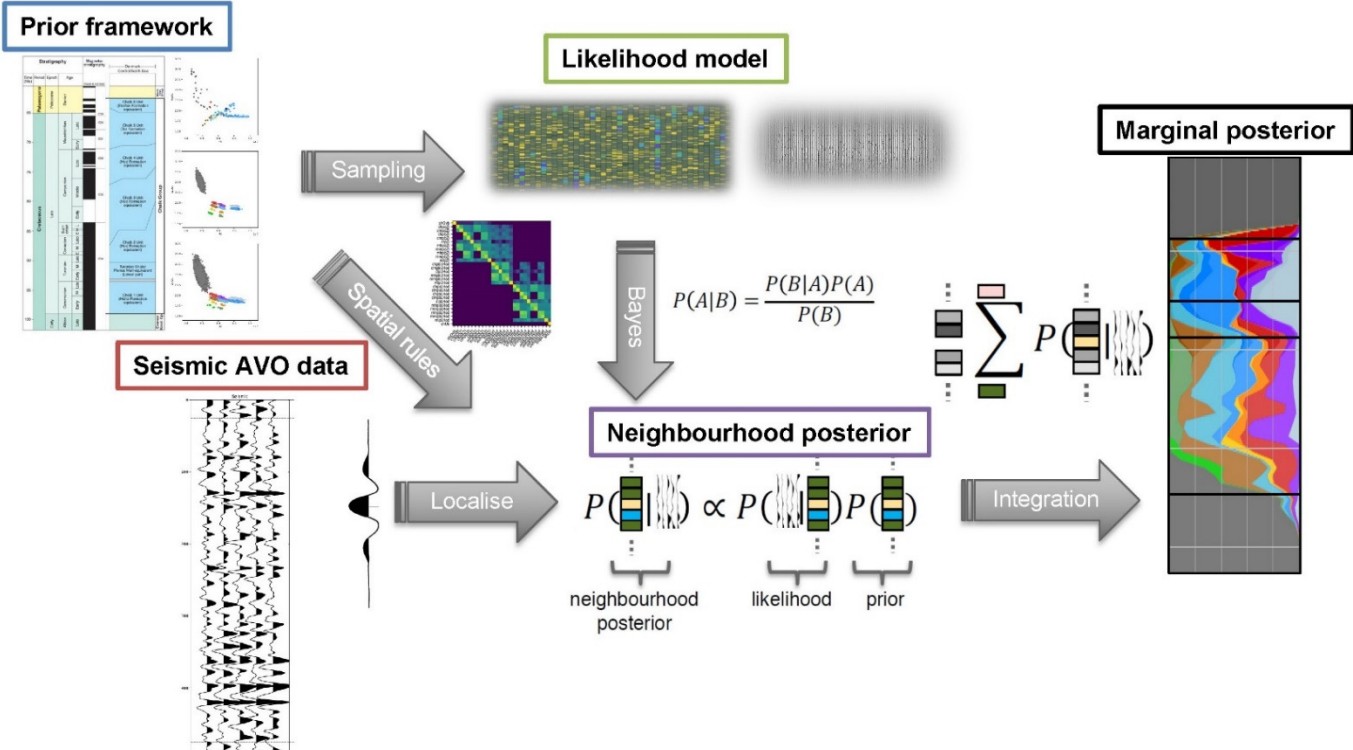

**Figure 3.** Schematic workflow of the DPI inversion (adapted from Goodway et al. [17]).

## 3. Study Area, Data Coverage and Reservoir Model

The wedged chalk play is a hydrocarbon prospect with the Tor and Ekofisk Formations as reservoir targets within the Upper Cretaceous and earliest Paleocene chalk successions, located 10 km northwest from the Siah-NE-1X well and approximately 5 km east from the Nora-1 well (Figure 4). It represents a wedge of potential high-porosity reservoir chalk between a lower low-porosity Hod Formation and upper Paleocene marlstone and clay (Figure 4). The reservoir facies onlap the northern flank of the Pollernes Ridge inversion structure, thereby resulting in a pinch-out in the southern direction (see Chalk Group thickness map in Figure 2). The main geological challenge that was investigated with the DPI tool is related to distinguishing the reservoir formation from the overburden and underburden formations, as well as better resolving the reservoir pinch-out.

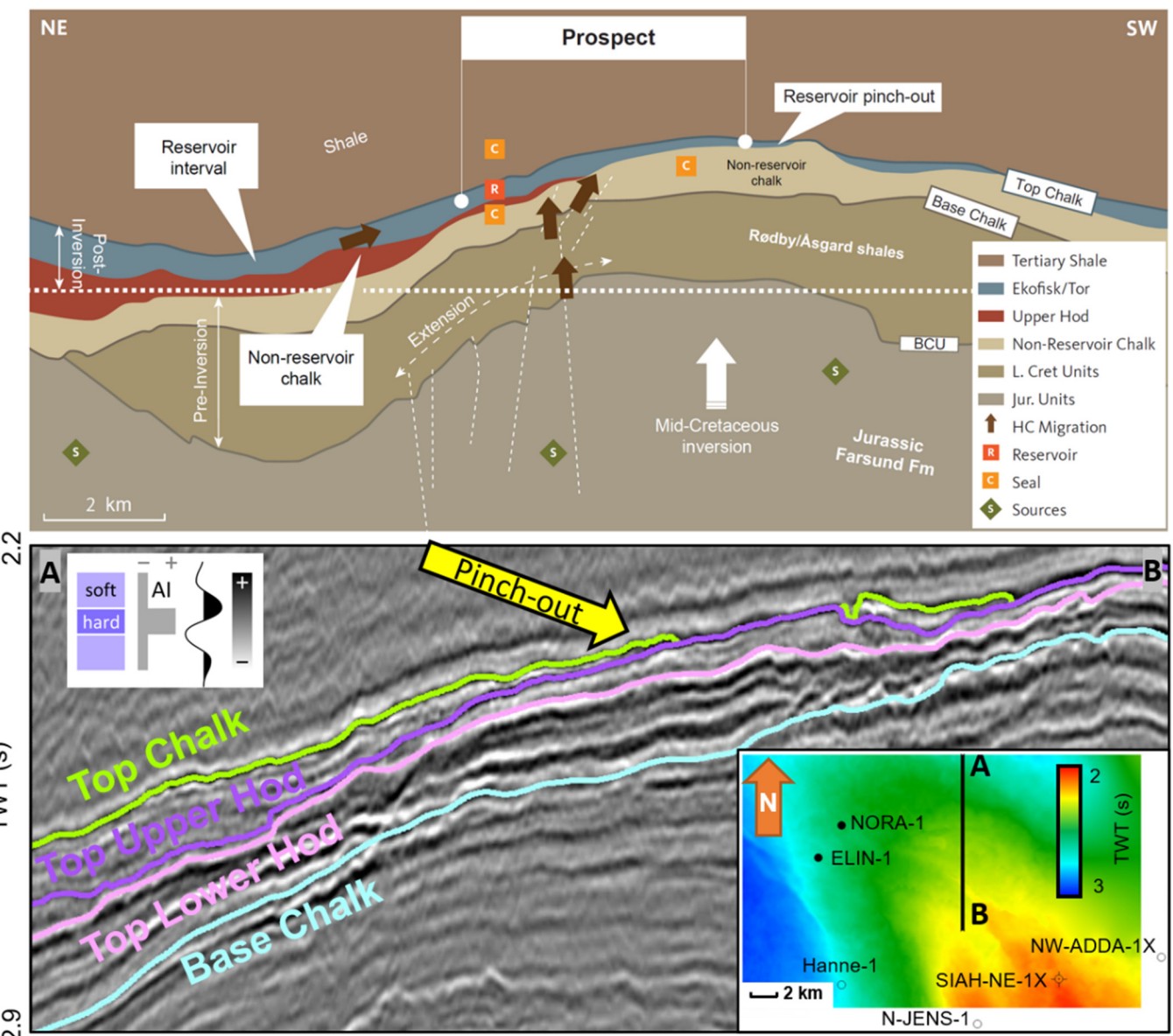

**Figure 4.** Upper: prospect cartoon section showing the main elements of the hydrocarbon chalk prospect. Tor and Ekofisk reservoir pinch out against the Top Upper Hod and are charged with hydrocarbons via faults. Lower: 2D seismic crossline section through the prospect area with the corresponding Top Chalk surface in MapView showing nearby wells. TWT: two-way time.

A static reservoir model of the chalk prospect was constructed by first performing 3D seismic interpretation of the main chalk events to map out the large-scale architecture of the Chalk Group (Table 1). The top and base of the Chalk Group form the top and base of the static model, while the Top Lower Hod forms the basal surface upon Upper Hod, Tor and Ekofisk formations are onlapping upon. The top Upper Hod marker forms the base of the porous reservoir chalk intervals, or top of the low-porosity sealing unit. The layering architecture is set such that it follows the seismic geometries (e.g., conformable, erosional and onlapping) and can be recognized in Figure 5. Four wells (Hanne-1, Elin-1, Nora-1 and Siah-NE-1X) were included in the static model to provide initial porosity distributions in three defined zones (Tor/Ekofisk, Hod and Hidra) and used to define the vertical variability of porosity (how rapidly the porosity change) (Figure 2 for well locations). Then a simplified porosity–depth dependency was implemented to account for increasing burial compaction with increasing depth, which is a simplification, since differences in overpressure could offset this trend, but it was decided for the current study to be an adequate approximation [18]. The trend surface was constructed by using the minimum and maximum burial difference of the three zones, and the maximum decrease in porosity (as a result of increasing burial depth) was calculated from the porosity–depth curves. For the Ekofisk-Tor, this difference was 12%; for Hod and Hidra, the difference was 13% between top of the static model and the deepest level of each zone.

**Table 1.** Seismic interpretation parameters.

| Name Seismic Marker | Architecture Element | Horizon Type |
| --- | --- | --- |
| Top Chalk Group/Top Ekofisk Fm. | Top Reservoir | Erosional |
| Top Upper Hod | Base Reservoir | Conformable |
| Top Lower Hod | Base onlapping surface | Conformable |
| Base Chalk Group/Base Hidra Fm. | Base lower seal | Base |

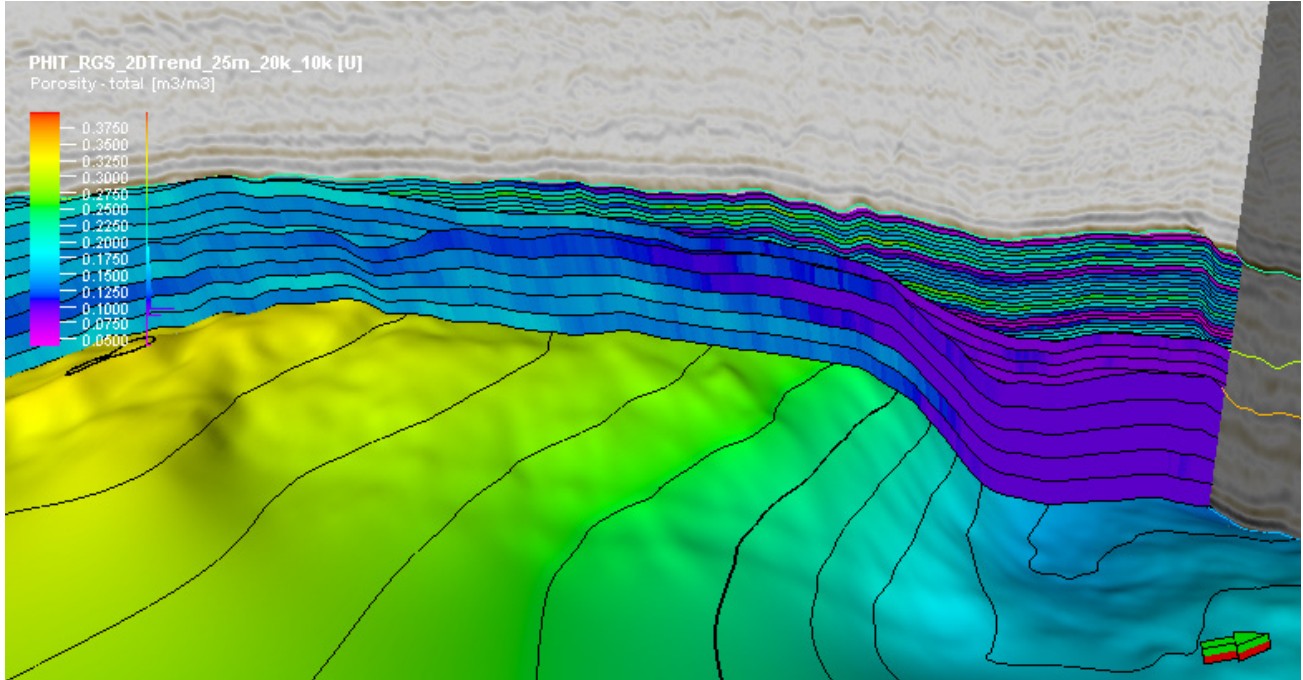

**Figure 5.** N–S extracted section through the 3D static model and the wedged chalk prospect, with modeled porosity values. Note that the reservoir units have a finer layering and the highest porosities, and they pinch out to the south. Note also the decrease in porosity in the deeper northern part of the model as a result of increased burial compaction.

## 4. Inversion Setup

Two different seismic inversion approaches were performed: (1) deterministic AVO inversion (the traditional approach) [19,20] and (2) the improved direct probabilistic inversion (DPI) approach [10,17,21]. The basic workflows for the inversions are outlined in Table 2.

**Table 2.** Inversion workflow.

| Preparation phase (input) | |
| --- | --- |
| Conditioning of input seismic CDP gathers<br>Angle stacking<br>Preparation of well database<br>Seismic well ties<br>Wavelet estimation | |
| **Inversion phase (Output)** | |
| Deterministic AVO inversion<br>*Output*:<br>Acoustic impedance (AI) and Vp/Vs<br>(elastic properties) | Direct probabilistic inversion (DPI)<br>*Output*:<br>Probabilities of facies classes: shale,<br>high-porosity chalk with oil, high-porosity<br>chalk with brine, medium-porosity chalk with<br>oil, medium-porosity chalk with brine,<br>low-porosity chalk |
| **Comparison phase** | |

The main challenge in the inversion workflow for the chalk prospect was related to the lack of nearby shear sonic log data to generate rock physics likelihood models (RPMs). NW-Adda-1X was the only well that contained a reliable shear sonic log through the Upper Cretaceous interval and was therefore used for this purpose, although it is located far away from the chalk prospect (Figure 2) and is 150 ms TWT shallower. The three closest wells (Siah-NE-1X, Nora-1 and Elin-1) were used for quality control of the DPI results against the available petrophysical log data (water saturation, volume of clay and porosity).

The first step for the DPI setup is to define the facies classes that we suspect can be present within the target chalk prospect interval. In this study, the facies were defined by petrophysical cutoffs within the following main zones:

- Shaly overburden (ShOvb);
- Ekofisk-Tor (Zone of interest: Zi);
- Upper Hod (UHod);
- Lower Hod (LHod);
- Shaly underburden (ShUb).

The facies definitions are shown by the color scheme in Figure 6. The defined set of facies classes include variations in porosity (high, medium and low), whether the chalks are clean or marly (tight and shaly chalks) and whether brine or hydrocarbon saturates the pore volume. For the low-porosity scenario, only a brine saturation scenario was considered. Each facies is subdivided in accordance with which zones the facies are mainly relevant for. This leaves us with 26 distinct facies, represented by each circle slice in Figure 6. For example, the Upper Hod (UHod) zone is expected to be brine-saturated chalks. Hence, the prior probability for oil is set close to zero, so the UHod is not included in any of the oil-saturated-facies scenarios. Constraining the number of facies from prior geological knowledge is important to reduce the non-uniqueness of the inverse problem.

An a priori probability model for the facies was defined from regional interpreted horizons, assuming equal proportions of the relevant facies classes inside the various zones. The a priori local spatial structure of the facies was formulated as a 1D Markov process [22] sampled from the facies thickness distributions from well observations (Figure 7).

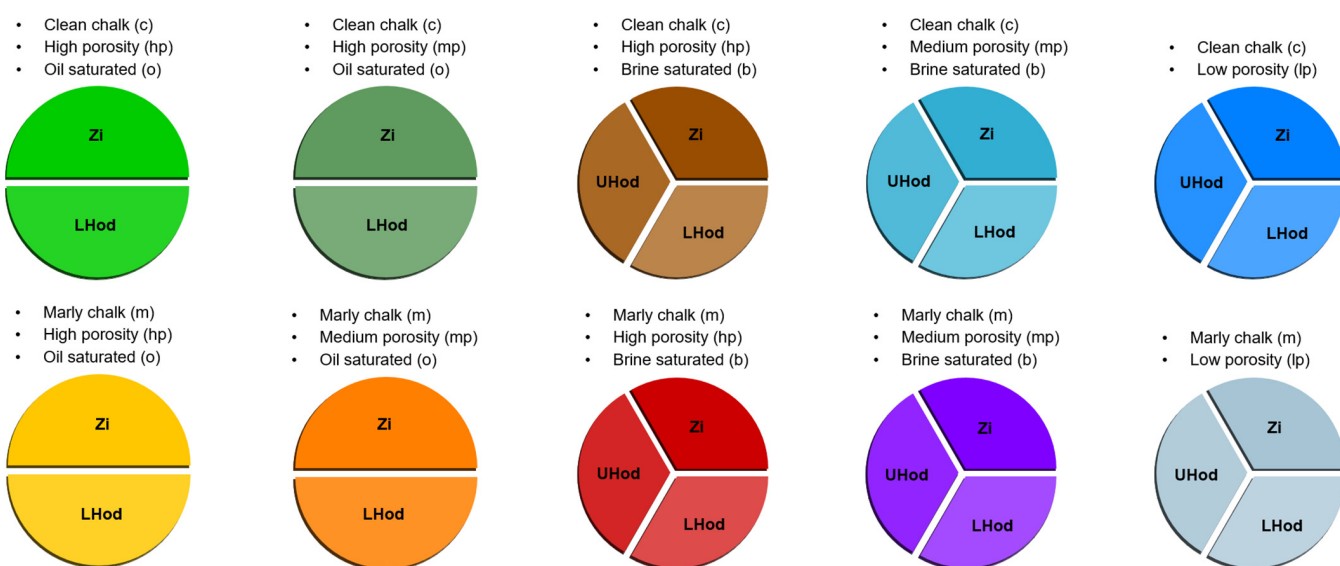

**Figure 6.** Facies class legend. The presented facies classes focus on the three main zones within the target interval: Ekofisk-Tor (Zone of interest: Zi), Upper Hod (UHod) and Lower Hod (LHod).

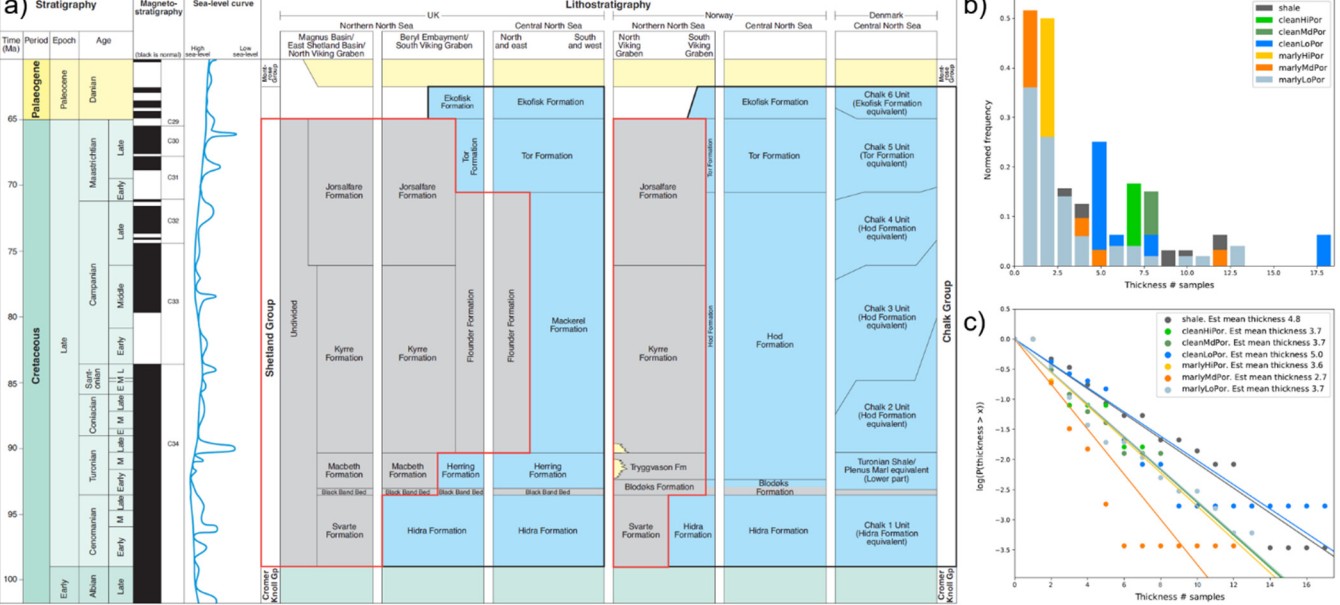

**Figure 7.** (**a**) Stratigraphic subdivision from Surlyk, et al. [23]. The target interval is within Late Cretaceous. (**b**) Histogram of observed facies thickness distribution. (**c**) Logarithm to the probability of thicknesses larger than a given thickness. The linear trend indicates that thicknesses are exponentially distributed in line with a vertical Markov spatial model [10].

Another important inversion setup is the definition of a robust prior probability model for the various facies classes. For example, if we anticipate high-porosity brine-saturated chalk to be more likely present within a specific depth interval, we assign a higher prior probability to that facies within that interval. For the vertical ordering (or stacking) of the facies classes in the prior model, we want the facies classes to obey some certain physical and geologically consistent rules. In this study, these were rules as follows:

- Thickness distributions are estimated to be exponential;
- Facies vertical ordering follows a first-order Markov process (Figure 7);
- The ordering statistics vary between all intervals;

- Fluid gravitational ordering is assumed;
- Older sequences are always located below younger sequences;
- The elastic properties (acoustic impedance, Vp/Vs and density) within each facies can be modeled with Gaussian distributions;
- The correlations of elastic properties within a given facies are modeled with an exponential correlation model.

Figure 8 shows the Markov model transition probability matrix from which short facies windows (typically 5 to 15 samples) are generated to build likelihood models. The matrix describes, in terms of probabilities, all the vertical combinations of facies classes within the various zones (Ekofisk-Tor, Upper Hod and Lower Hod) in accordance with the listed rules. The matrix represents the probability of a facies (row index) at a given location having facies (column index) in the sample below. Notice that the observed facies classes are chronologically defined along each axis, starting with the uppermost shale overburden (shOvb) in the first row/column and then stepwise through all the facies scenarios (Figure 6) toward the lowermost shale underburden (shUb). For example, a zero-transition probability in row chpbZi (water-saturated highly porous clean chalk) and column chpoZi (oil-saturated highly porous clean chalk) is due to the fact that water is not allowed to occur above oil (Figure 6). The corresponding prior probability models are shown for the Siah-NE-1X, Nora-1 and Elin-1 wells in Figure 9. Notice also that smooth prior probability transitions at the horizons' surfaces are due to their assigned uncertainties.

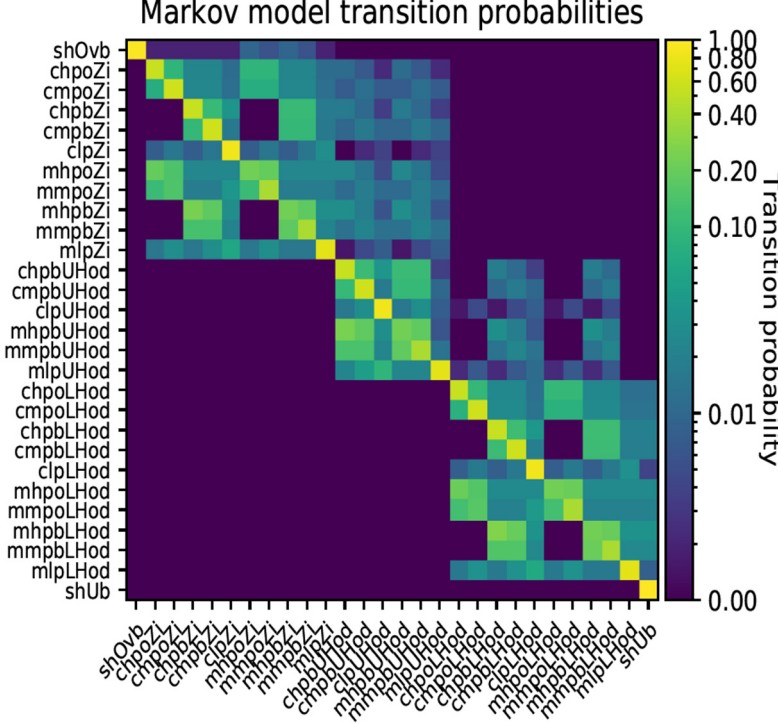

**Figure 8.** Markov model transition probability matrix. Probabilities are given in volume fractions (*v/v*); shOvb and shUb are shale overburden and underburden, respectively. The information in the matrix is used locally when generating facies windows for generating likelihood models.

Figure 10a shows resampled AI vs. Vp/Vs data from the NW-Adda-1X well within the target interval, and Figure 10b shows the corresponding statistical rock physics models (RPMs) as dashed ellipses (Gaussian probability density functions) [8,24]. The RPMs extends upon the observed distribution of elastic properties in each facies within reasonable ranges to incorporate plausible facies variations. Notice that the in situ data in the target interval do not represent all the facies classes that we want to investigate. For example, because the NW-Adda-1X is a dry well, a Gassmann fluid substitution [25] was performed

to model different oil saturated scenarios in different porosity (high, medium and low) and lithology (clean and marly) settings, as well. In addition, some of the RPMs that were underrepresented in the NW-Adda-1X data were edited to give them more meaningful elastic properties in accordance with established rock physics models [26]. Notice the overlap of the various RPMs (Figure 10b). Sufficient separation between the RPMs is crucial for the inversion to be able to distinguish between the corresponding facies classes when only weak spatial information exists, as it is linked to the seismic expression (i.e., amplitudes). Therefore, the more overlap we observe between two different RPMs in Figure 10b, the more difficult it is for the inversion to discriminate between them if there is not prior strong spatial information.

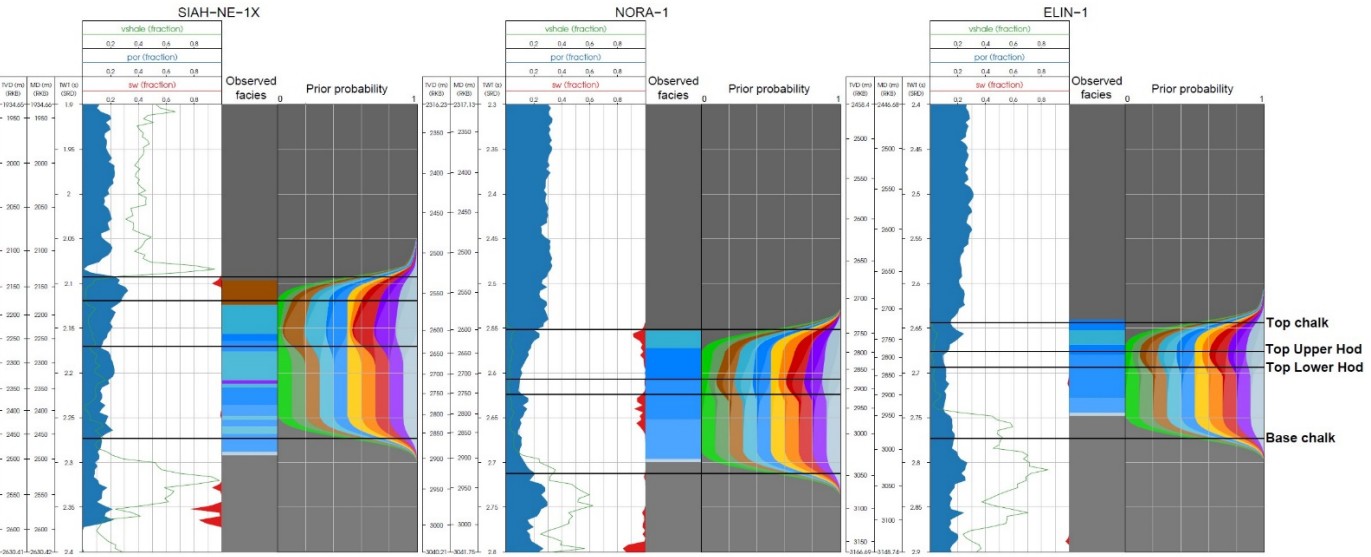

**Figure 9.** Prior model setup for the Siah-NE-1X (**left**), Nora-1 (**middle**) and Elin-1 (**right**) wells. For each well from left-to-right, shale volume (vclay), porosity (por) and water saturation logs; observed (or reference) facies profile; and the prior probabilities as a function of time/depth. The four seismic surfaces are plotted: Top Chalk, Top Upper Hod, Top Lower Hod and Base Chalk.

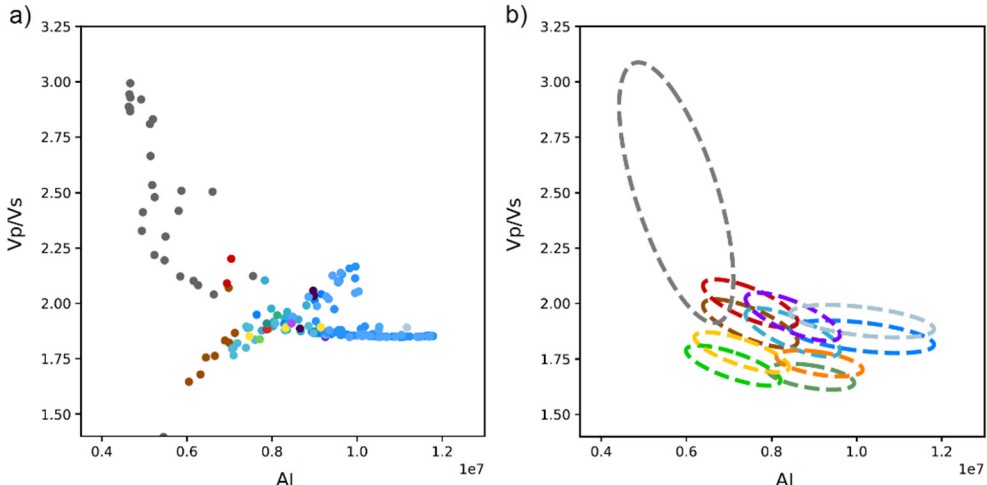

**Figure 10.** Acoustic impedance (AI) versus Vp/Vs: (**a**) data from the NW-Adda-1X that contain shear sonic measurements; (**b**) the corresponding rock physics model likelihoods (RPMs) defined from the data and fluid substitution modeling. The large gray ellipse represents the over- and underburden RPMs. Refer to Figure 6 for the facies colors.

To approach RPMs at a seismic scale, some realizations were performed based on the well log elastic data. The standard deviation of the Gaussian RPMs (Figure 10b) is approached by matching the observed and realized reflectivities. Figure 11 shows the estimated reflectivity distributions for the acoustic impedance (AI) and Vp/Vs to the left from the NW-Adda-1X data in the top row and the corresponding cross-plot to the right. The middle and lower rows represent Gaussian simulated data for the various facies classes' testing variance factors of 0.3 and 0.5, respectively. The histograms from using a variance factor of 0.3 (middle row) seem to better match the estimated reflectivities from the well (upper row). In this way, AI and Vp/Vs data are simulated based on the statistical information from the NW-Adda-1X well data to populate the dataset used for defining the RPMs.

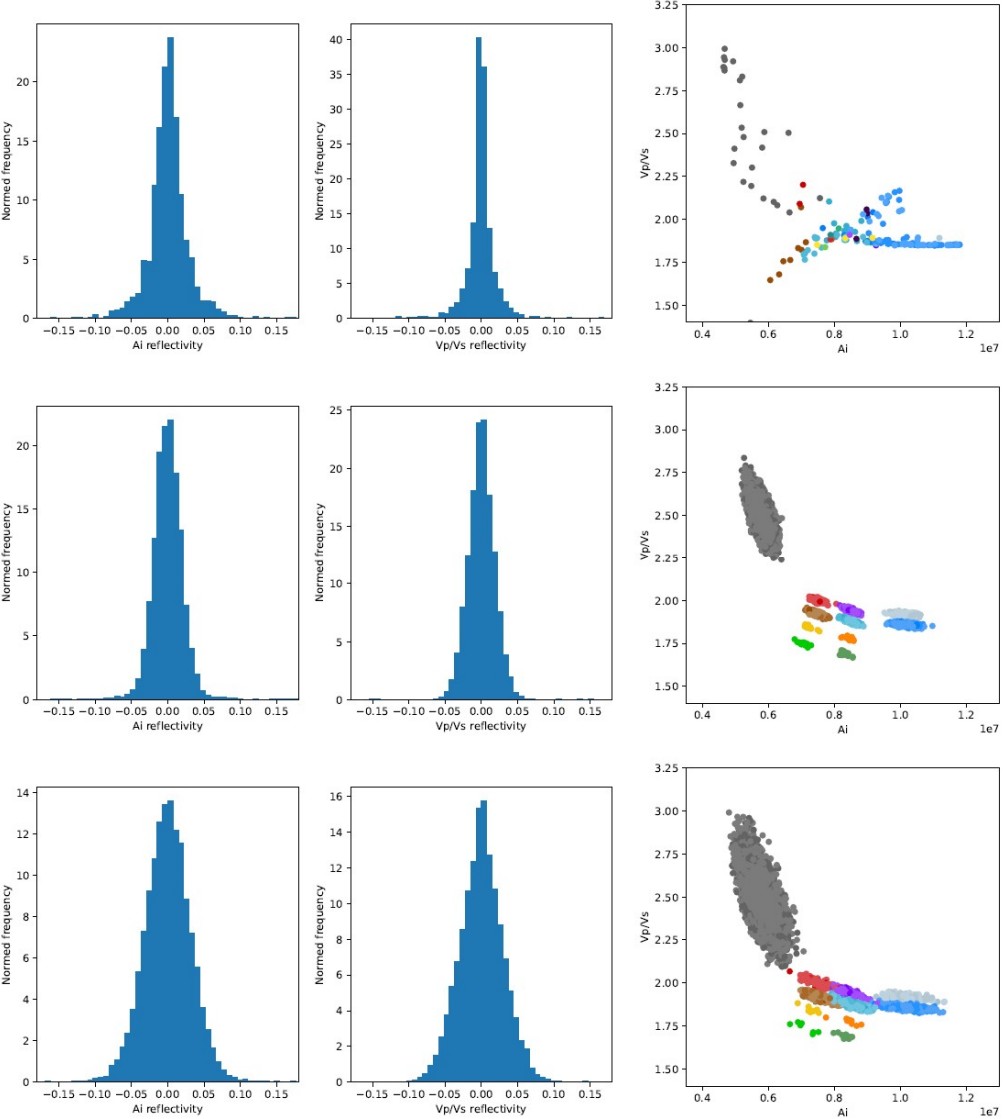

**Figure 11.** Realizations of elastic AI and Vp/Vs to approach rock physics models at seismic scale. Upper row: estimated reflectivities and well data from NW-Adda-1X. Middle and lower rows: simulated reflectivities and elastic data. Refer to Figure 6 for the facies colors.

Statistical wavelets for seven partial angle stacks (0–40°) were extracted from the seismic (Figure 12). These were used together with the Aki and Richards [15] AVO forward model and an uncorrelated noise model.

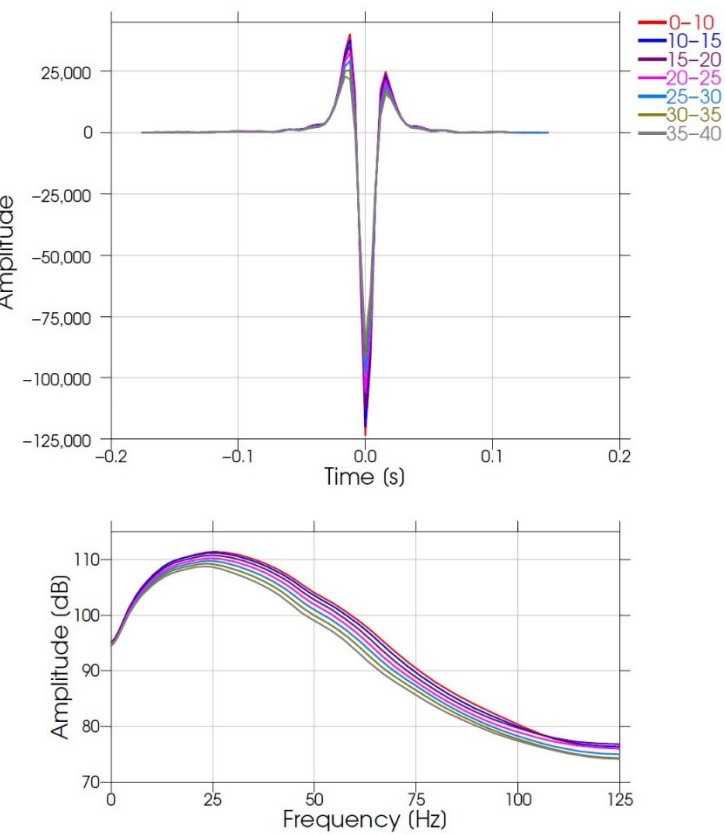

**Figure 12.** Statistical wavelets extracted from seven partial angle stacks extracted from a fixed time window around the Upper Cretaceous.

## 5. Inversion Results

Figure 13 shows a seismic composite line intersecting the Siah-NE-1X, Elin-1 and Nora-1 wells and through the prospect area in the north. Figures 14 and 15 show the deterministic AVO inversion results for AI and Vp/Vs, respectively, and Figure 16 shows the corresponding DPI results. Both inversions used similar estimated wavelets and horizon surfaces as input. The DPI results showed that the facies' inverted for is geologically consistent and in line with geological expectations and thereby reduced geological uncertainties associated with the chalk prospect. The high-porosity brine features seen toward the outermost left of the section (Figure 16) were difficult to interpret both on the seismic (Figure 13) and using the traditional deterministic AVO inversion approach (Figures 14 and 15).

In Figure 17, the posterior probabilities are shown for the wells. Notice how the prior probabilities in Figure 9 were transformed to posterior by using the DPI. In general, if the prior and posterior probabilities are similar, the seismic AVO data do not contain any useful information about the various facies classes. In this case, however, the significant deviation between the prior and posterior probabilities in each of the wells implies that the seismic AVO data drive the inversion. Comparing the inversion results with the observed facies in each well, we see that the classification seems to distinguish the clean and marly chalks reasonably well, whereas it misclassifies some higher-probability anomalies for hydrocarbons in Siah-NE-1X and Nora-1 in the Lower Hod Formation. However, given that the inversion relies on limited and distant well-log data from NW-Adda-1X (Figure 10) and the prominent overlap of the RPMs (Figure 10b), the results are acceptable.

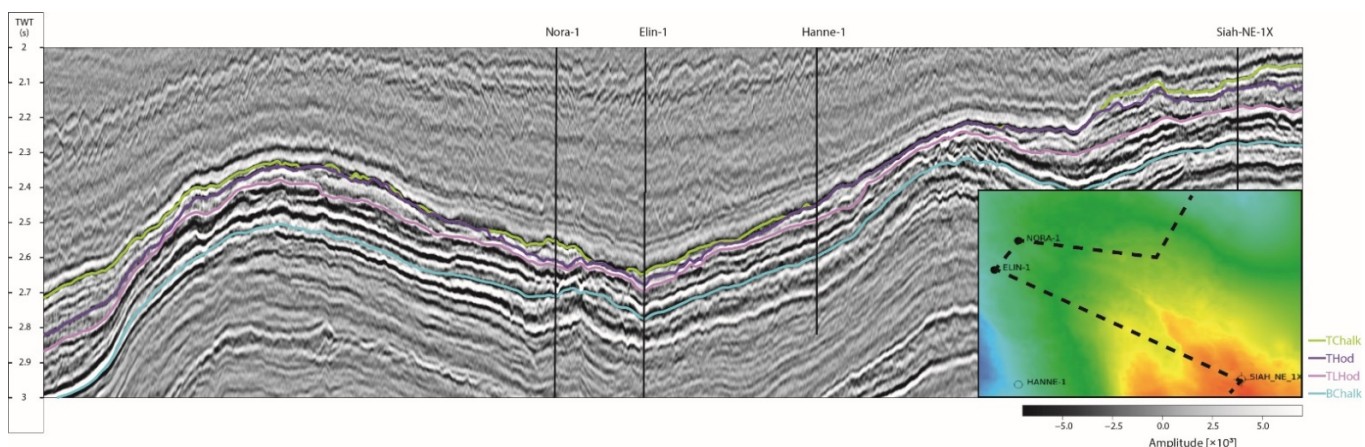

**Figure 13.** The 10–15° angle-stack seismic data from a composite seismic line through the nearby wells and the wedged chalk prospect (dashed black line in MapView). Well trajectories intersected by the composite line are plotted. The four seismic surfaces used are plotted in different colors.

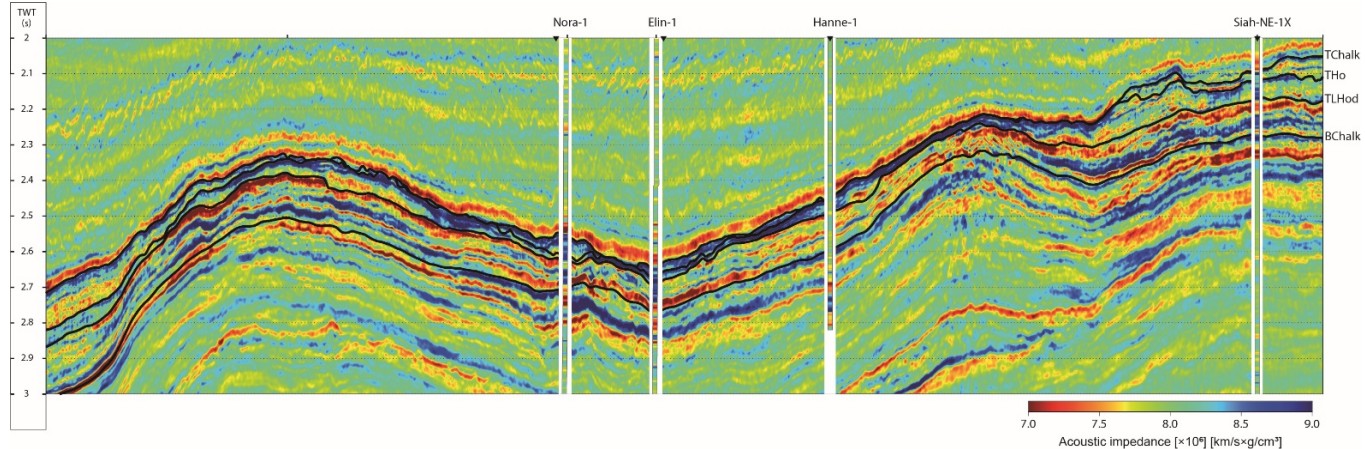

**Figure 14.** Deterministic inversion result—acoustic impedance (AI) along the composite seismic line. Well trajectories intersected by the composite line are plotted with AI log data where available. The four seismic surfaces used are plotted in black.

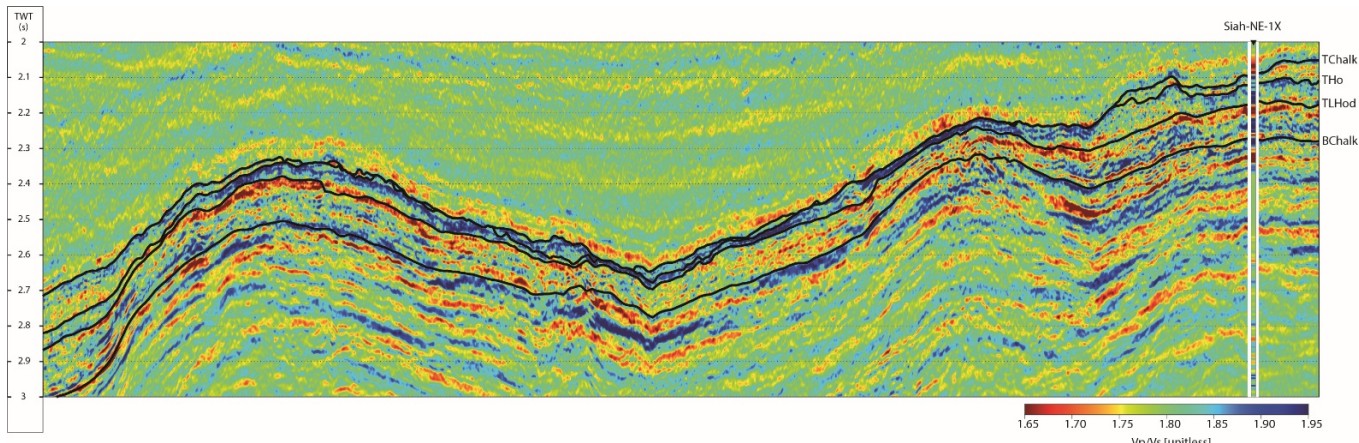

**Figure 15.** Deterministic inversion result—Vp/Vs ratio from a composite seismic line. Well trajectories intersected by the composite line are plotted with Vp/Vs log data where available. The four seismic surfaces used are plotted in black.

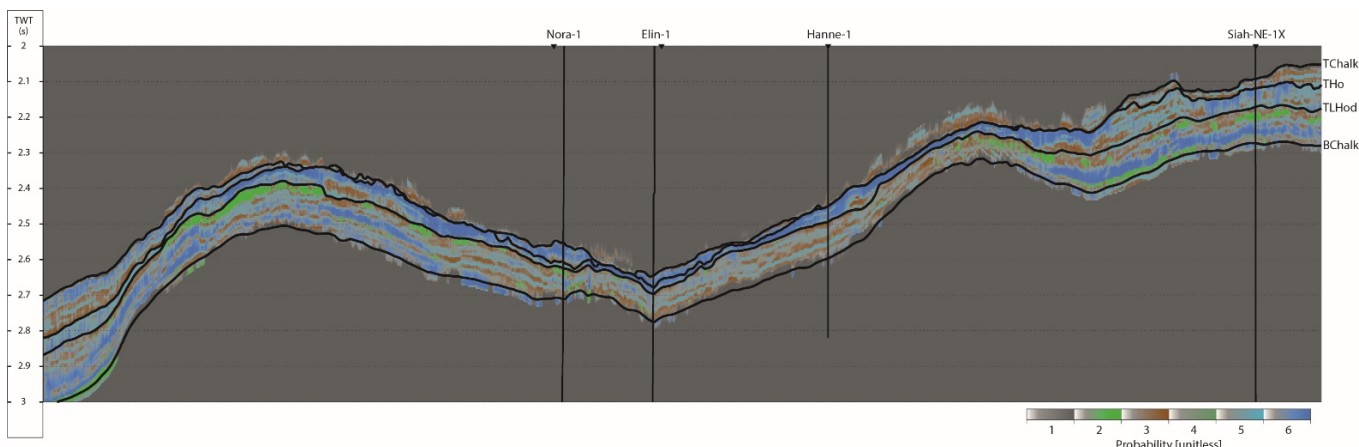

**Figure 16.** Direct probabilistic inversion result along the composite line. The individual colors represent the most likely geological facies' inverted for (see Figure 6) and is from left to right: (1) shale, (2) high-porosity chalk with oil, (3) high-porosity chalk with brine, (4) medium-porosity chalk with oil, (5) medium-porosity chalk with brine and (6) low-porosity chalk. Well trajectories intersected by the composite line are displayed as black lines. The four seismic surfaces used are plotted in black.

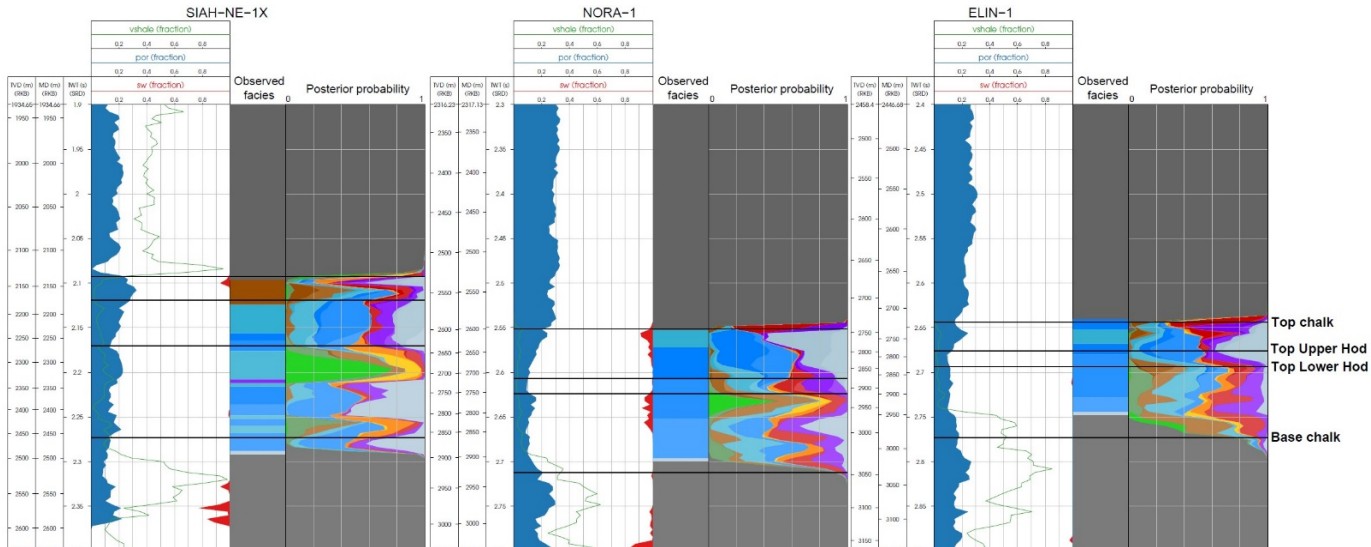

**Figure 17.** Inversion results for the Siah-NE-1X (**left**), Nora-1 (**middle**) and Elin-1 (**right**) wells. For each well, from left-to-right, shale volume (vclay), porosity (por) and water saturation logs; observed (or reference) facies profile; and the posterior probabilities as a function of time/depth. The four seismic surfaces are plotted: Top Chalk, Top Upper Hod, Top Lower Hod and Base Chalk.

The interesting element of the DPI results are the prediction of potential highly porous reservoir chalks that are onlapping the inversion structure and therewith also the pinch-out position, which affects the potential in situ hydrocarbon volumes. The traditional reservoir modeling workflow uses internal chalk surfaces, geostatistical extrapolation of log data and a simple porosity–depth relationship to obtain the porosity distribution and pinch-out position (Figure 18a). The DPI results show the probability of high-porosity chalks (Figure 18b), and it looks fairly similar to the distribution from the geostatistical reservoir modeling workflow. The main difference between these two approaches is that the DPI results rely on actual remote seismic measurements covering the prospect area, in contrast to the geostatistical modeling that is exclusively based on the propagation of well log data, using specific variogram settings (e.g., until what lateral and vertical distance are porosity

values still related to one another). The DPI results are therefore more data-driven than the geostatistical model and can be used to refine the static reservoir model to better reflect the actual seismic data.

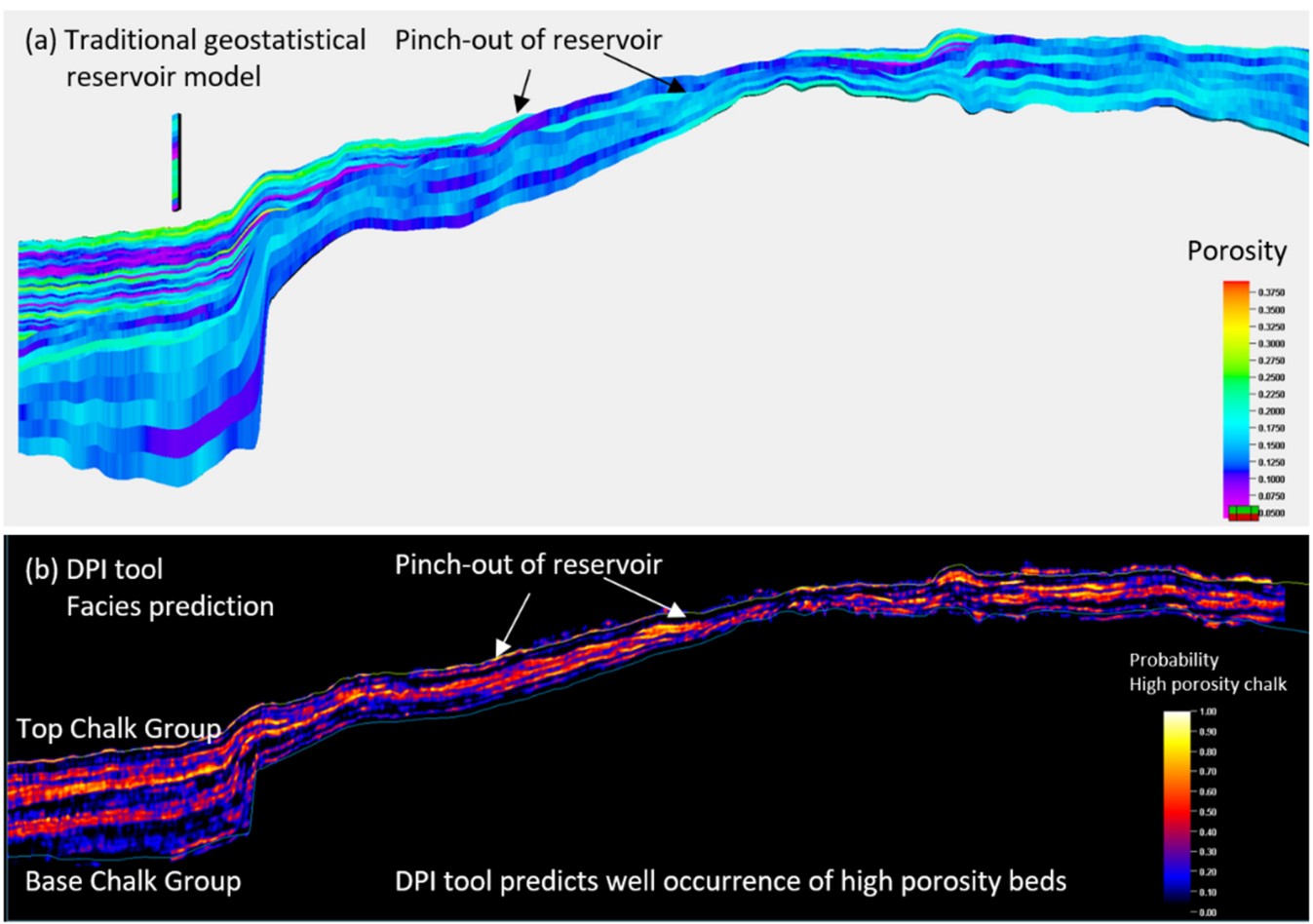

**Figure 18.** Comparison of (**a**) traditional geostatistical reservoir modeling method, extrapolating data from well logs; and (**b**) prediction of high-porosity chalk facies from the DPI tool.

## 6. Implementing DPI Results into Reservoir Modeling

There are different applications for the results obtained from the DPI tool that are useful in the reservoir modeling/characterization workflow (Figure 19). For each defined facies, the DPI tool takes in a prior probability 3D model and transforms it to a posterior probability 3D volume that can aid reservoir characterization in multiple ways. First, high-probability anomalies for specific facies (for example of high porous chalks) can guide the interpretation of boundary/interfaces of various reservoir zones, for example, by adjusting the reservoir top or base horizon to better match the high-probability anomalies from the DPI results and, thus, the structural framework. This can be particularly useful in areas where it is challenging to accurately map specific reservoir targets with conventional seismic interpretation techniques, for example, due to limited seismic resolution in the data. Secondly, by populating the reservoir zones with depositional facies, the 3D probability volumes of each facies can be inserted as a 3D probability constraint in geostatistical methods (sequential indicator simulation or Truncated Gaussian simulation). Another possibility is to extract 2D probability maps for each facies in specific zones and insert those as 2D constraints in the same geostatistical methods. An alternative avenue is to use the Most Probable Facies 3D volume, which is constructed from selecting the facies exhibiting the highest posterior probability at each point within the 3D volume (Figure 19). This volume contains discrete facies definitions that can directly be used to constrain the static

reservoir model. Another important contribution from the inversion is the higher resolution obtained for the various facies classes that can resolve thin layers below tuning thickness to obtain a more accurate and detailed reservoir model. Thus, these workflows allow for the better delineation and definition of reservoir zones and geometry and improved construction of 3D facies models through seismic constraints obtained from the DPI tool.

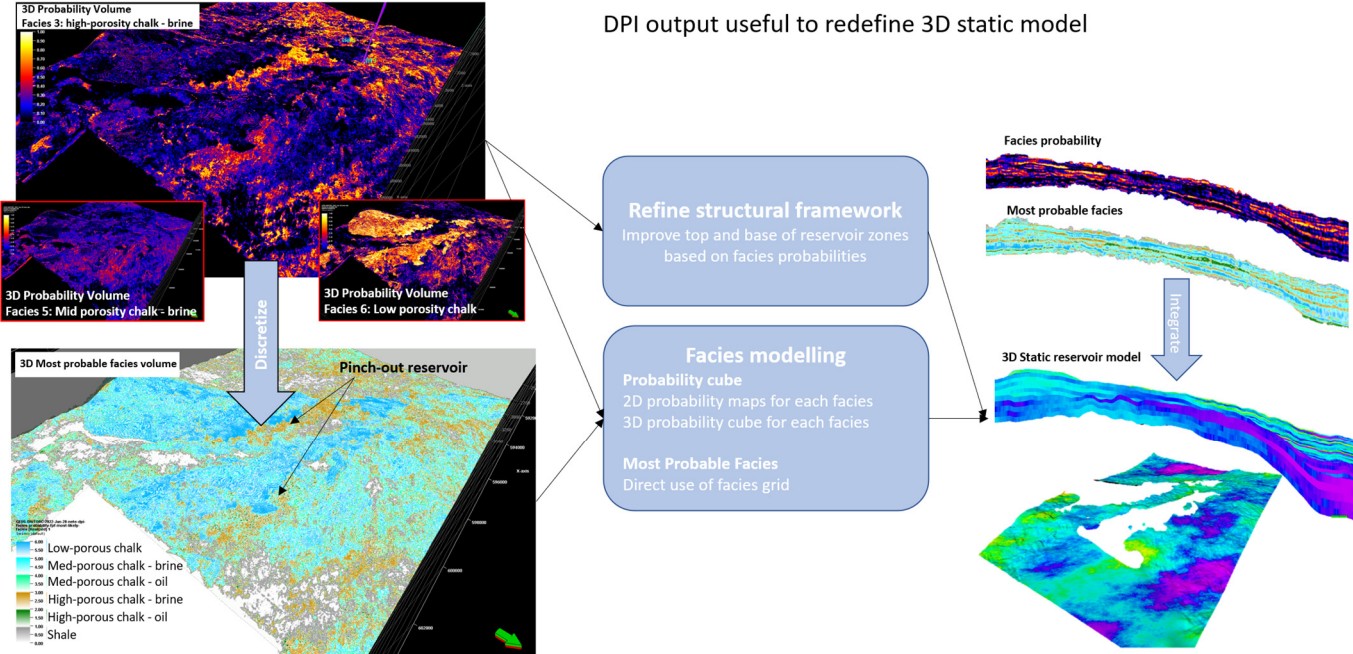

**Figure 19.** Output of the DPI tool for reservoir modeling/characterization purposes.

## 7. Conclusions

A novel direct probabilistic inversion using seismic pre-stack data was formulated as a Bayesian inference problem and demonstrated on a wedged chalk reservoir prospect on the Upper Cretaceous Chalk Group in the Danish North Sea. The results provided additional reservoir information in a probabilistic manner that was challenging to resolve based on more traditional workflows, such as conventional seismic interpretation, deterministic seismic inversion and geostatistical reservoir modeling. Although nearby well log data were limited, using more distant well data gave a more quantitative seismic interpretation useful for prospect derisking. The quantitative accuracy and potential of the inversion is proportional to the availability and quality of both seismic and well data, although this study also demonstrates the usefulness in frontier exploration settings as a reconnaissance tool. The direct probabilistic inversion tool can also be used for other applications, such as screening for geological $CO_2$ or hydrogen storage sites or geothermal resources.

**Author Contributions:** K.B., conceptualization (equal), investigation (supporting), methodology (supporting), validation (supporting), project administration (lead), writing—original draft (lead), and writing—review and editing (lead); I.H., conceptualization (equal), investigation (lead), methodology (equal), software (equal), validation (lead), writing—original draft (supporting), and writing—review and editing (supporting); F.S., conceptualization (equal), investigation (supporting), validation (equal), writing—original draft (supporting), and writing—review and editing (supporting); A.F.J., conceptualization (equal), investigation (supporting), methodology (lead), software (lead), validation (supporting), writing—original draft (supporting), and writing—review and editing (supporting); P.F., conceptualization (equal), investigation (supporting), validation (supporting), and writing—original draft (support); A.B., conceptualization (equal), investigation (supporting), validation (supporting), project administration (equal), and writing—original draft (supporting). All authors have read and agreed to the published version of the manuscript.

**Funding:** This research was funded by the Energy Technology Development and Demonstration Program (EUDP), grant number 64018-0591.

**Data Availability Statement:** The data that support the findings of this study are not publicly available. Data are, however, available from the authors upon reasonable request and with permission of GEUS.

**Conflicts of Interest:** The authors declare no conflict of interest.

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
