# Peer review of "Characterizing a Wedged Chalk Prospect in the Danish Central Graben Using Direct Probabilistic Inversion"

_geosciences, doi:10.3390/geosciences12050194_

Round 1

Reviewer 1 Report

This paper describes the use of direct probabilistic inversion method using seismic pre-stack data as input to characterize a wedged chalk reservoir prospect.

First of all I would like to congratule the authors for the excellent work!

Despite I consider that the paper could be accepted in the present form, I would like the authors to clarify some doubts and consider including some more information:

  • In the abstract, the authors state: "The objective was to better resolve the lateral extent", but the markov chain process is applied in a trace-by-trace base and no lateral continuity information is considered in the process. This also lead to the lack of "regularization" in the inversion result and some artifacts are shown, rigth?
  • I would like to see some information about  the computational cost of the method (time, computer configuration, warm-up and total cycles of the MCMC process).

Author Response

Dear reviewer,

Thank you for your comments on our paper. Here is our responses to your comments:

  1. "The objective was to better resolve the lateral extent", but the markov chain process is applied in a trace-by-trace base and no lateral continuity information is considered in the process. This also lead to the lack of "regularization" in the inversion result and some artifacts are shown, rigth?"

    Response:
    In areas with wells and good geological understanding we have strong vertical prior information. Getting good horizontal prior information (connectivity) is difficult to get and requires outcrop studies etc. Bayesian methods (like DPI) allows to infuse soft and strong vertical constraints/prior information into the problem making it possible in some cases to resolve below tuning. By treating each angle gather trace as a separate piece of data (1D) any horizontal connectivity we see are coming from data and not from any added prior idea of horizontal information. Using only vertical prior information we believe it is a more conservative choice and practically our clients typically have limited if any horizontal prior information. DPI is fundamentally not limited to 1D and to squeeze better (more certain) results out a full 3D would be better IF you trust your horizontal prior information and the 1D convolutional model (ideally use point spread functions).

    Artifacts can have many causes: poor DPI approximation (see below), edge effects around zone of interest since we are not adequately modelling over- and underburden facies sequence and/or elastic properties - requires modeling at least half a wavelet length below and above zone of interest. The major cause of artifacts is varying SNR. So "over fitting" where SNR is poor - for example where the convolutional model is poor. Notice that DPI does not rely on the convolutional model as forward model and any model can be used that takes more of the wavefield into account (for proof of concept see https://www.earthdoc.org/content/papers/10.3997/2214-4609.202037034). 

  2. "I would like to see some information about  the computational cost of the method (time, computer configuration, warm-up and total cycles of the MCMC process)."

    Response:
    The DPI method is not a MCMC method but is in the family of approximative Bayesian methods (MCMC being an exact method in infinite steps). MCMC is not feasible on commercial sized projects with millions of traces. DPI is not sampling traces of or generating "pseudo-traces" since that is not feasible except for very small seismic areas and a very limited number of facies. In DPI the same mathematical operation is carried out at each seismic grid point - fast analytical evaluation of many thousands likelihoods. DPI is achieving speed by localizing and approximating the seismic likelihood with a multivariate gaussian. This makes it possible to "compare" (weight) many neighborhood realizations ("windows") in the facies domain and find the pointwise probability using Monte Carlo integration at a given point in the seismic grid. DPI can to some extent work with large facies windows by conditional sampling which significantly improves the quality of the approximation by including more of the spatial statistics of the prior. The number of samples needed for stable results increase with size of the window and can easily be QC'ed.  Some Bayesian methods generate an exhaustive list of combinations which explode with increased window size/many facies.

    Run time of DPI on a single trace with 6-8 angle stacks is typically around 0.1 to 2.0 seconds depending on details seen in many commercial projects (depend on spatial length scales of facies, wavelet, facies complexity etc.).  Routinely we do ~10 million trace projects with 6 to 8 stacks sometimes with multiple azimuths (16 stacks).

Reviewer 2 Report

This paper presents a novel and interesting approach of using Markov process and Bayesian inference to solve the geophysical inversion problem. The data example has demonstrated the ability in better classifying different litho-fluid facies, as well as high resolution in delineating the boundary and pinch-out of a chalk prospect reservoir.

Author Response

We thank Reviewer 2 for the positive response to our paper.